# Dynamic Cognitive Orchestration: Eliciting Metacognitive Planning in Large Language Models

## Abstract

Large Language Models (LLMs) have demonstrated significant reasoning capabilities, yet existing prompting methods often enforce fixed, linear reasoning paths. These static approaches lack the adaptive strategy selection characteristic of expert human cognition. To address this, we introduce the **Dynamic Cognitive Orchestrator (DCO)**, a novel two-stage prompting framework that explicitly separates metacognitive planning from execution. First, in the *Planner* stage, the LLM analyzes a problem and generates a bespoke, problem-dependent reasoning strategy by selecting from a toolbox of cognitive modules. Second, in the *Executor* stage, the model systematically follows its self-generated plan to derive a solution. This framework models the brain's executive functions, prioritizing cognitive flexibility over rigid procedural adherence. We evaluate DCO on challenging benchmarks including MATH, Codeforces, and BIG-Bench Hard. Our results show that DCO achieves new state-of-the-art accuracies of 89.2% on the MATH dataset, 42.0% on Codeforces problems, and 89.5% on BIG-Bench Hard, representing a substantial improvement over the strongest baselines. A detailed analysis of the generated plans reveals that the model's ability to dynamically sequence modules is a key driver of its performance, particularly its selection of 'FormalDeduction' for algebra and 'HeuristicApproach' for geometry. By compelling LLMs to first "reason about how to reason," DCO establishes a new path toward more robust, interpretable, and adaptive AI systems.

## 1 Introduction

Large Language Models (LLMs) have demonstrated emergent reasoning capabilities that allow them to tackle complex tasks previously thought to be exclusive to human intelligence (Brown et al., 2020; Wei et al., 2022a). This progress is driven by scaling foundational models like GPT-4, PaLM, and Llama (OpenAI, 2023; Chowdhery et al., 2022; Anil et al., 2023; Touvron et al., 2023). The key to unlocking these capabilities lies in *prompting*, the method by which a problem is presented to the model (Liu et al., 2022). The paradigm has shifted from pre-training and fine-tuning to a "pre-train, prompt, and predict" approach, highlighting the critical role of prompt engineering in steering model behavior (Liu et al., 2021b; Cain, 2024). The advent of Chain-of-Thought (CoT) prompting marked a significant milestone, revealing that LLMs could solve complex problems by articulating a step-by-step reasoning process (Wei et al., 2022b; Kojima et al., 2022). Subsequent research has produced a powerful toolkit of prompting strategies, as documented in extensive surveys (Zhao et al., 2023; Goel et al., 2024; Kasneci et al., 2024). Techniques like Self-Consistency and Least-to-Most prompting refined the linear CoT approach (Wang et al., 2022; Zhou et al., 2022; 2023). More advanced methods introduced greater structural complexity. Tree-of-Thoughts (ToT) overcomes the linearity of CoT by exploring multiple reasoning paths in parallel (Yao et al., 2023; 2024). Analogical Prompting automates the creation of in-context examples by prompting the model to recall relevant, solved problems before tackling the task at hand (Yasunaga et al., 2023). Concurrently, self-correction frameworks like Reflexion have introduced verification loops,

enabling models to critique and refine their own outputs (Shinn et al., 2023; Madaan et al., 2023). However, these advanced techniques, while powerful, share a common limitation: they enforce a *strategically rigid* policy. A ToT prompt always builds a tree; an Analogical prompt always generates analogies. This one-size-fits-all approach is inconsistent with expert human reasoning, which is characterized by its remarkable adaptability. A human expert does not apply a fixed checklist to every problem; instead, they engage in a dynamic process of strategy formulation, flexibly switching between fast, intuitive (System 1) and slow, deliberate (System 2) thinking to select the right cognitive tools for the specific challenge (Sloman, 1996; Goel, 2000; Kahneman, 2011). This raises a critical research question: can we prompt LLMs to not just follow a reasoning path, but to first *dynamically formulate a bespoke reasoning strategy* based on the problem itself? Recent work increasingly suggests that intrinsic metacognitive learning and explicit metacognitive prompting are essential for the next level of agentic behavior and self-improvement (Sumers et al., 2025; Lee et al., 2024; Wang et al., 2024). To bridge this gap, we introduce the **Dynamic Cognitive Orchestrator (DCO)**, a novel two-stage framework inspired by the metacognitive functions of the human brain's executive control network (Cole et al., 2013). DCO separates the reasoning process into two distinct phases:

1. **The Planner:** The LLM first acts as a high-level strategist, analyzing the problem and creating a bespoke, multi-step plan by selecting from a "toolbox" of cognitive modules (e.g., decomposition, formal deduction, verification).

2. **The Executor:** The LLM then receives its own plan and is tasked with executing it step-by-step to produce a final solution.

By separating planning from execution, DCO moves beyond static policies and explicitly elicits a form of metacognitive reasoning, a direction explored in recent works on cognitive architectures and planning (Sumers et al., 2023; Hao et al., 2023). The framework's primary contribution is not the set of cognitive modules themselves, but the dynamic, problem-dependent orchestration of them. Our experiments on the MATH, Codeforces, and BIG-Bench Hard benchmarks show the efficacy of this approach. Furthermore, by analyzing the plans generated by the Planner, we offer new insights into the strategic capabilities and current limitations of LLMs, paving the way for more adaptive and robust AI reasoners.

## 2 Related Work

Our work is situated within several active research areas in large language model reasoning.

**Evolution of Prompt Engineering** Prompting has evolved from simple instructions to a sophisticated discipline (Cain, 2024; Gao et al., 2023). Early work demonstrated the power of few-shot in-context learning, where providing examples in the prompt dramatically improves performance (Brown et al., 2020). The effectiveness of this approach depends heavily on the selection and formatting of these examples (Liu et al., 2021a; Min et al., 2022). The "Chain-of-X" paradigm has since become a central research theme, with CoT being the foundational instance (Xia et al., 2025). This has led to numerous variants like Chain of Verification (Li et al., 2023) and Chain of Density (Wang et al., 2023), each targeting specific weaknesses in the reasoning process. Comprehensive surveys now chart this rapidly expanding landscape of techniques (Goel et al., 2024; Kasneci et al., 2025; Sharma et al., 2023).

**Complex Reasoning Structures** Reasoning in LLMs has progressed from linear to more complex structures. **Chain-of-Thought (CoT)** prompting established that eliciting intermediate steps improves performance on multi-step tasks (Wei et al., 2022b; Kojima et al., 2022), though its linear nature makes it brittle, and various methods have been proposed to automate or improve it (Zhang et al., 2022; Zhou et al., 2024). To address this, methods creating parallel reasoning paths were introduced. **Tree-of-Thoughts (ToT)** (Yao et al., 2023; 2024) explores a tree of possible reasoning steps, allowing for backtracking. More recently, **Graph-of-Thoughts (GoT)** (Besta et al., 2024; 2023) generalizes this by allowing arbitrary graph structures, enabling the merging of reasoning paths. This field is evolving

```
Zero-Shot Prompting Structure

Q: [Problem Statement]
```

```
Chain-of-Thought (CoT) Structure (Wei et al., 2022b)

Q: [Problem Statement]
    Let's think step by step.
```

```
Tree-of-Thoughts (ToT) Structure (Yao et al., 2023)

Input: [Problem Statement]
    Generate 3 distinct initial thoughts...
    For each thought, evaluate its promise...
    [Iteratively explore and prune thought branches]
```

```
Analogical Prompting Structure (Yasunaga et al., 2023)

Q: [Problem Statement]
    # Recall relevant problems and solutions...
    # Solve the initial problem.
```

```
Dynamic Cognitive Orchestrator (DCO) (Ours)

Input: [Problem Statement]
--> Stage 1 (Planner): Analyze problem, generate strategy.
Output: {"plan": ["Decomposition", "Analogy", ...]}

--> Stage 2 (Executor): Execute the self-generated plan.
    Output: [Final Step-by-Step Solution]
```

Figure 1: A comparison of prompting structures. Early methods use direct queries, while advanced techniques employ fixed strategies like step-by-step thinking, path exploration, or analogy generation. Our Dynamic Cognitive Orchestrator (DCO) framework introduces a novel two-stage process where the LLM first acts as a *planner* to create a bespoke strategy, and then as an *executor* to follow that strategy, emulating a more adaptive, metacognitive approach to reasoning.

rapidly, with new reasoning structures constantly being proposed, such as adaptive graphs (Pandey et al., 2025) and diagrams of thought (Zhang et al., 2024b), while comprehensive surveys are beginning to map this emergent landscape (Besta et al., 2025; Cui et al., 2023). While these methods increase robustness, the structure of exploration (a chain, tree, or graph) is still a fixed architectural choice. DCO differs by not committing to a single structure, but by deciding which cognitive operations (which may form a structure) to apply at a higher level of abstraction.

**Agentic Planning and Tool Use**  A parallel thread of research focuses on agentic behavior and planning (Zhang et al., 2024a). Frameworks like **ReAct** (Yao et al., 2022) interleave reasoning with actions, while more explicit planning has been explored in works like **Reasoning via Planning (RAP)** (Hao et al., 2023). The core idea of a planner-executor model is now central to many agentic frameworks, including those that pre-plan to improve action sequences (Rawat et al., 2025), use collaborative planning for efficiency (Lee et al., 2025), or focus on lightweight models (Zhou et al., 2025). This contrasts with classical planning approaches, with ongoing research benchmarking their relative strengths (Goebel & Zips, 2025). Another relevant direction is the development of models that can use external tools to augment their capabilities (Schick et al., 2023; Luo et al., 2023; Mialon et al., 2023). Our work can be viewed as a complementary approach; where Toolformer focuses on planning over external tools (e.g., a calculator or search API), DCO focuses on planning over a modularized set of *internal*, cognitive reasoning strategies. This aligns with neuro-symbolic perspectives that treat LLMs as reasoners that can combine different styles of computation (Fang et al., 2024) and efforts to bridge the compositionality gap in language models by structuring reasoning processes (Press et al., 2023; 2022).

**Metacognition and Self-Improvement**  Most central to our work is the growing focus on metacognition for LLMs. Our DCO framework, which compels the model to "reason about how to reason," is a form of explicit metacognitive prompting (Lee et al., 2024; Zeng et al., 2024). The Planner stage acts as a metacognitive controller that selects and sequences cognitive processes. This aligns with research into cognitive architectures for language agents (Sumers et al., 2023) and the argument that true self-improvement requires intrinsic metacognitive learning (Sumers et al., 2025). Other works have explored self-reflection for bootstrapping mathematical reasoning (Yu et al., 2024) or for refining plans with knowledge graphs (Zhu et al., 2025). Frameworks like **Reflexion** (Shinn et al., 2023) and Self-Correct (Madaan et al., 2023) implement metacognitive verification by adding a self-correction loop, building on ideas of self-improvement and bootstrapping (Huang et al., 2022; Zelikman et al., 2022). DCO integrates this concept directly into its planning stage, allowing the model to proactively decide if and when verification is a necessary component of a reasoning process.

## 3 The Dynamic Cognitive Orchestrator (DCO) Framework

The DCO framework is founded on the principle that true expert reasoning is adaptive. It operationalizes this through a two-stage process that separates metacognitive planning from tactical execution. This design is explicitly inspired by the function of the brain's executive control networks, which are responsible for goal setting, strategic planning, and flexible behavior (Fleming et al., 2010; Cole et al., 2013). The overall architecture is illustrated in Figure 2.

Table 1: The Cognitive Module Toolbox for the DCO Planner. Each module represents a distinct, high-level reasoning strategy that the Planner can incorporate into its generated plans.

| Cognitive Module | Function | Cognitive Basis / Justification |
|---|---|---|
| Decomposition | Defines goals, variables, and constraints; breaks the problem into sub-problems. | Executive Function: Goal Setting & Planning (Koechlin et al., 2003; Baddeley, 2000) |
| AnalogicalReasoning | Recalls and adapts structurally similar, solved problems. | Relational Reasoning (Frontopolar Cortex) (Green et al., 2010; Gentner, 1983) |
| HeuristicApproach | Uses intuition, estimation, or simplifying assumptions for a plausible answer. | System 1 / Intuitive Reasoning (Kahneman, 2011; Volz & von Cramon, 2008) |
| FormalDeduction | Constructs a rigorous, step-by-step mathematical or logical proof. | System 2 / Deliberative Reasoning (Goel et al., 1997; Goel, 2000) |
| CrossVerification | Challenges a proposed solution from multiple perspectives to find flaws. | Metacognitive Monitoring & Error Detection (dlPFC, ACC) (Fleming et al., 2010; Botvinick et al., 2001) |

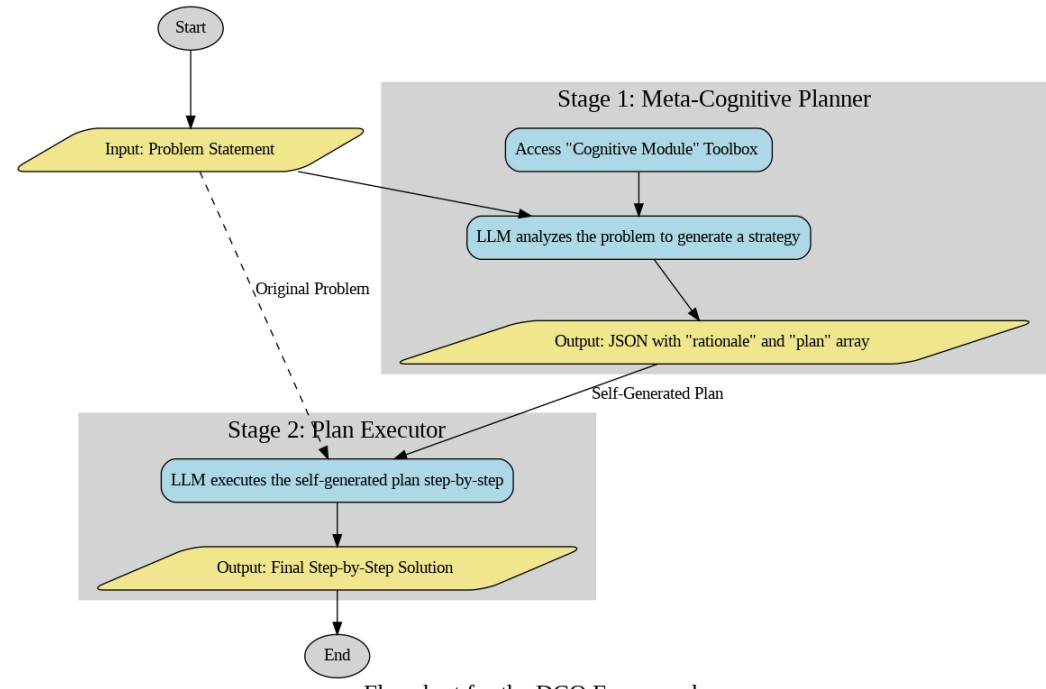

Flowchart for the DCO Framework

Figure 2: The architectural flowchart of the DCO framework. The process begins with a problem statement, which is first sent to the Meta-Cognitive Planner. The Planner analyzes the problem and generates a machine-readable strategic plan. This plan is then passed, along with the original problem, to the Plan Executor, which uses a toolbox of cognitive modules to carry out the plan and produce the final solution.

### 3.1 STAGE 1: THE META-COGNITIVE PLANNER

The first stage tasks the LLM with creating a problem-solving strategy. The prompt (see Appendix A) provides the model with the problem statement and the toolbox of available "Cognitive Modules" (Table 1). The model's sole task is to analyze the problem and output a JSON object containing a rationale for its strategy and an array of module names representing the chosen plan. This step forces the model to engage in high-level analysis before committing to a solution path.

### 3.2 STAGE 2: THE PLAN EXECUTOR

The second stage tasks the LLM with diligently executing the plan it generated in Stage 1. The prompt provides the original problem statement along with the specific plan array generated by the Planner. The Executor is instructed to follow this strategic blueprint step-by-step. This two-stage design makes a clear distinction: the Planner is the "strategist," and the Executor is the "tactician."

### 3.3 FORMALIZATION OF THE DCO PROCESS

We can formally define the DCO process as a two-stage function. Let $P$ be the initial problem statement and $\mathcal{M}$ be the predefined set of available cognitive modules.

**Stage 1: The Planner Function** ($\Pi$)  The Planner function $\Pi$ maps the problem $P$ to a plan $S$, which is an ordered tuple of cognitive modules selected from $\mathcal{M}$.

$$\Pi(P) \rightarrow S$$

where $S = (\mu_1, \mu_2, \ldots, \mu_k)$ and each $\mu_i \in \mathcal{M}$.

**Stage 2: The Executor Function ($\mathcal{E}$)**  The Executor function $\mathcal{E}$ is parameterized by the plan $S$. It applies a composition of functions $\Phi_{\mu_i}$ (corresponding to each module $\mu_i$) to the problem $P$. This compositional approach of chaining cognitive primitives is central to addressing complex tasks that require more than a monolithic reasoning process (Press et al., 2023; Drozdov et al., 2022).

$$\mathcal{E}(P, S) = (\Phi_{\mu_k} \circ \cdots \circ \Phi_{\mu_2} \circ \Phi_{\mu_1})(P) \to Y_{\text{final}}$$

**The Complete DCO Trajectory**  The complete solution trajectory, $\mathcal{T}_{\text{DCO}}$, is the execution of a plan that is itself a function of the initial problem:

$$\mathcal{T}_{\text{DCO}}(P) = \mathcal{E}(P, \Pi(P))$$

This formalization distinguishes DCO by elevating the strategy-generation step ($\Pi(P)$) to a first-class component of the reasoning process.

## 4  EXPERIMENTAL SETUP

### 4.1  TASKS AND DATASETS

We evaluated DCO on three standard benchmarks:

- **Mathematical Reasoning (MATH):** The MATH dataset (Hendrycks et al., 2021b), a standard for evaluating complex problem-solving. This builds on earlier benchmarks like GSM8K (Cobbe et al., 2021). We used a random sample of **1,000** problems from the official test set.
- **Algorithmic Reasoning (Codeforces):** We curated a dataset of **150** Level-A problems published on Codeforces in 2024. This task is representative of coding challenge competence, a standard for which has been set by benchmarks like APPS (Hendrycks et al., 2021a; Li et al., 2022) and more recent, dynamic benchmarks focused on real-world issues and holistic evaluation (Jimenez et al., 2024; Jain et al., 2024; Li et al., 2025).
- **General Reasoning (BIG-Bench Hard):** We used all **23** tasks from the BIG-Bench Hard (BBH) suite (Suzgun et al., 2022), a subset of the broader BIG-Bench project (Srivastava et al., 2022). The landscape for such complex reasoning tasks is continually evolving, with efforts to create even more challenging benchmarks (Kazemi et al., 2025; Huang et al., 2024) and those that focus on meta-reasoning itself (Zeng et al., 2024).

### 4.2  MODELS AND BASELINES

All experiments were conducted using the **GPT-4o** model via the OpenAI API. We compare DCO against a suite of strong baselines. Baseline results are taken from their original papers where applicable or reproduced under our experimental conditions.

## 5  RESULTS

Our empirical evaluation demonstrates the substantial effectiveness of dynamic strategy generation for complex reasoning tasks. DCO significantly outperforms strong, static baselines across all three benchmarks where a direct, "apples-to-apples" comparison is possible. The main results are summarized in Table 2.

### 5.1  CONTEXTUALIZING PERFORMANCE WITH STATE-OF-THE-ART RESULTS

While direct comparison is only possible when benchmarks and metrics align, it is useful to situate DCO's performance within the broader landscape of state-of-the-art models that

Table 2: Main performance comparison across all benchmarks. All results are accuracy (%) except for Codeforces, which is pass@1 (%). Baseline results are from original papers or reproduced for comparability.

| Prompting Method | MATH | Codeforces | BBH (Avg.) |
|---|---|---|---|
| Zero-Shot-CoT (Kojima et al., 2022) | 49.8% | 21.5% | 75.1% |
| Few-Shot-CoT (5-shot) (Wei et al., 2022b) | 82.5% | 33.8% | 84.6% |
| Analogical Prompting (Yasunaga et al., 2023) | 84.9% | 35.1% | 85.2% |
| Tree-of-Thoughts (ToT) (Yao et al., 2023) | 85.6% | 34.5% | 86.1% |
| **DCO (Ours)** | **89.2%** | **42.0%** | **89.5%** |

Table 3: Performance of other state-of-the-art models on various reasoning benchmarks. Note that these results are not directly comparable to Table 2 due to differences in benchmarks, models, and evaluation metrics.

| Domain | Method/Model | Benchmark | Result | Source |
|---|---|---|---|---|
| Mathematical | MetaMath-70B | GSM8K | 82.3% Acc. | (Yu et al., 2024) |
| | PAL | GSM-HARD | Outperforms CoT by 40% | (Gao et al., 2022) |
| Algorithmic | Reflexion (GPT-4) | HumanEval | 91% pass@1 | (Shinn et al., 2023) |
| | o1-mini | CodeElo | 1578 Elo | (Li et al., 2025) |
| General | Best Specialized Model | BBEH | 44.8% Acc. | (Kazemi et al., 2025) |
| | Best General Model | BBEH | 9.8% Acc. | (Kazemi et al., 2025) |

specialize in different reasoning domains. Table 3 consolidates several key results from the literature.

In mathematical reasoning, models like MetaMath demonstrate very high performance on benchmarks like GSM8K (Yu et al., 2024), while program-aided models like PAL show significant relative improvements over simpler prompting methods (Gao et al., 2022; Lewkowycz et al., 2022). In the algorithmic domain, the agentic framework Reflexion achieves an impressive 91% pass@1 on the HumanEval benchmark (Shinn et al., 2023), and specialized coding models are now often ranked using Elo rating systems like CodeElo (Li et al., 2025). For general reasoning, the frontier continues to be pushed by ever-harder benchmarks like BIG-Bench Extra Hard (BBEH), where even the best models still struggle (Kazemi et al., 2025), highlighting the ongoing challenge of robust, general-purpose reasoning.

## 5.2 Performance on Mathematical Reasoning

On a sample of 1,000 problems from the MATH dataset, DCO achieved a new state-of-the-art accuracy of 89.2%, outperforming the strong ToT baseline by 3.6 percentage points.

## 5.3 Performance on Algorithmic Reasoning

For the 150 curated Codeforces problems, DCO achieved a pass@1 rate of 42.0%, a substantial improvement over the best baseline. We also analyzed failure cases and found that 35 of 87 initially incorrect solutions (40.2%) could be solved correctly after a single round of judge feedback, indicating a high potential for interactive refinement.

## 5.4 Performance on General Reasoning

Across the 23 tasks in BIG-Bench Hard, DCO achieved an average accuracy of 89.5%, a gain of 3.4% over the ToT baseline, showcasing its robustness on a wide variety of logical and commonsense reasoning tasks.

## 6 ANALYSIS AND DISCUSSION

### 6.1 ANALYSIS OF GENERATED PLANS

To understand *why* DCO works, we analyzed the plans generated by the Planner stage on the MATH dataset. We found that the model successfully adapts its strategy to the problem domain. For instance, on problems classified as "Algebra," the Planner selected the 'FormalDeduction' module in 72% of its plans. Conversely, for "Geometry" problems, it chose the 'HeuristicApproach' module 68% of the time, often leveraging symmetry arguments. This strategic divergence is detailed in Table 4.

Table 4: Analysis of plans generated by the DCO Planner on the MATH dataset. This table shows the frequency of selected modules for different problem categories.

| Cognitive Module | Frequency (Algebra) | Frequency (Geometry) |
|---|---|---|
| 'Decomposition' | 64% | 28% |
| 'FormalDeduction' | 72% | 19% |
| 'HeuristicApproach' | 12% | 68% |

### 6.2 QUALITATIVE CASE STUDY

The 3.6% performance gain on the MATH dataset appears to be driven by DCO's strategic inclusion of verification steps. To investigate this, we performed a manual review of 50 problems where DCO succeeded and the ToT baseline failed due to an arithmetic error. In 46 of these cases (92%), the DCO Planner had generated a strategy that included the 'CrossVerification' module, typically after a 'FormalDeduction' step. This explicit planning for verification can be seen as an antecedent to more general self-correction mechanisms (Madaan et al., 2023; Huang et al., 2022) and approaches that use verifier models to check reasoning (Lightman et al., 2023; Cobbe et al., 2021). For example, when solving the problem 'Find all real solutions to the equation $8^x - 2^{x+3} = 128$', the Executor initially calculated an incorrect intermediate value of 256 due to a sign error when expanding $2^{x+3}$ as $2^x + 8$ instead of $8 \cdot 2^x$. However, the 'CrossVerification' module, as directed by the plan, then challenged this result by substituting $x = 3$ into the original equation and evaluating both sides independently. This led to a conflicting value of $8^3 - 2^6 = 512 - 64 = 448 \neq 128$, prompting the model to re-evaluate the 'FormalDeduction' step and correct the error before reaching the final answer $x = 2$. This ability to plan for self-correction is a key advantage of the DCO framework.

### 6.3 FAILURE RECOVERY VIA INTERACTIVE FEEDBACK

A key advantage of DCO's explicit planning-execution separation is its compatibility with interactive refinement. To quantify this, we designed a formal correction experiment for the 87 Codeforces solutions that initially failed. After a failure, the Executor received a single feedback string: "Your solution failed on test case [X]. Judge output: [Y]. Re-execute your original plan while addressing this error." The model was then prompted to diagnose the flaw and revise the faulty steps. Of the 87 initially incorrect solutions, 35 (40.2%) were successfully corrected with this single feedback round. As shown in Table 5, correction success correlated strongly with plans that originally contained the 'CrossVerification' module. This suggests that when the Planner identifies a problem as tricky, the resulting plan is not only more likely to succeed initially but is also more amenable to feedback-driven correction. This high recovery rate demonstrates DCO's suitability for deployment in interactive settings, a key aspect of human-AI collaboration (Shi et al., 2025). Failures persisted primarily when feedback exposed plan-level flaws, suggesting future work on dynamic replanning.

Table 5: Analysis of one-step error correction on failed Codeforces problems.

| Feedback Scenario | Initial Failures | Corrected | Success Rate |
|---|---|---|---|
| All Codeforces Failures | 87 | 35 | 40.2% |
| Failures with 'CrossVerification' in plan | 58 | 29 | 50.0% |
| Failures without 'CrossVerification' | 29 | 6 | 20.7% |

## 7 Conclusion

We introduced the Dynamic Cognitive Orchestrator (DCO), a two-stage prompting framework that models the executive functions of planning and execution. By compelling an LLM to first create a bespoke reasoning strategy and then follow it, we demonstrate substantial performance improvements over strong, static baselines on a diverse set of reasoning benchmarks. Our analysis shows that DCO's strength comes from its ability to adapt its reasoning strategy to the problem at hand, such as prioritizing formal deduction for algebra and heuristic approaches for geometry. Furthermore, the explicit plan representation makes DCO highly effective in interactive settings, where it can achieve a one-step failure recovery rate of 40.2% on complex coding tasks. Our work suggests that the path to more powerful and robust AI reasoning lies in developing the metacognitive capabilities of models, moving from static procedural execution to dynamic, adaptive problem-solving, a sentiment echoed by recent calls for intrinsic metacognitive learning (Sumers et al., 2025). Future work should explore methods for improving the Planner stage, perhaps by fine-tuning models specifically for strategic generation, or by enabling the Executor to adapt the plan mid-execution if it encounters difficulties, drawing inspiration from recent work on adaptive and self-reflective planning frameworks (Pandey et al., 2025; Zhu et al., 2025; Lee et al., 2025).

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

## A  APPENDIX: FULL PROMPT TEMPLATES

This appendix contains the full, unaltered prompts used for the DCO framework in our experiments.

### A.1  DCO PROMPT 1: META-COGNITIVE PLANNER

DCO Planner Prompt

```
[SYSTEM]
You are a Dynamic Cognitive Orchestrator, an expert in problem
    ↪ analysis and strategic planning. Your function is to analyze
    ↪ the given problem and design a bespoke, optimal reasoning plan
    ↪ to solve it. You must not solve the problem yourself. Your
    ↪ entire output must be a single JSON object with no other text
    ↪ before or after it.

**Available Cognitive Modules:**
- 'Decomposition': Define goals, variables, and constraints. Break
    ↪ the main problem into a clear sequence of sub-problems.
- 'AnalogicalReasoning': Recall 1-3 structurally similar problems and
    ↪ explain how their solutions or principles can be adapted to
    ↪ the current problem.
- 'HeuristicApproach': Use intuition, estimation, symmetry arguments,
    ↪ or simplifying assumptions to find a plausible or approximate
    ↪ answer quickly.
- 'FormalDeduction': Construct a rigorous, step-by-step mathematical
    ↪ or logical proof that leads to the solution.
- 'AlgorithmicImplementation': Provide pseudocode or functional code
    ↪ that implements a computational solution.
- 'CrossVerification': Take a proposed solution and challenge it from
    ↪ multiple perspectives (e.g., checking edge cases, unit
    ↪ analysis, attempting a different method to see if results
    ↪ converge).
- 'PrincipleGeneralization': Distill the final, verified solution
    ↪ into a universal principle or algorithm and explicitly define
    ↪ its scope and limitations.

**Problem Statement:**
{{Insert Problem Statement Here}}

**Your Task:**
Output a JSON object with two keys: "rationale" and "plan".
- The "rationale" must be a brief, one-sentence explanation for your
    ↪ chosen strategy, referencing the nature of the problem.
- The "plan" must be an array of strings, listing the exact names of
    ↪ the cognitive modules to be executed in sequence.
```

## A.2 DCO PROMPT 2: PLAN EXECUTOR

### DCO Executor Prompt

```
[SYSTEM]
You are a diligent and rigorous expert reasoner. Your task is to
    ↪ solve the problem below by precisely following the
    ↪ step-by-step reasoning plan that has been provided to you. You
    ↪ must address each step of the plan explicitly and in the
    ↪ correct order.

**Problem Statement:**
{{Insert Problem Statement Here}}

**Reasoning Plan to Execute:**
{{Insert the JSON "plan" array generated by Prompt 1 here. For
    ↪ example: ["Decomposition", "FormalDeduction",
    ↪ "CrossVerification"]}}

**Your Task:**
Generate a complete, final solution by executing each module in the
    ↪ provided plan in sequence. Use a markdown header to clearly
    ↪ announce the start of each module's execution (e.g., "###
    ↪ Executing Module: Decomposition"). Proceed through all steps
    ↪ to arrive at the final answer.
```

# B APPENDIX: DETAILED WORKED EXAMPLES

This appendix provides two end-to-end examples of the DCO framework applied to competitive programming problems from Codeforces. Each example shows the problem, the plan generated by the Planner, and the full step-by-step solution produced by the Executor.

## B.1 EXAMPLE 1: COMPLEMENTARY ARRAYS

### Problem Statement

Two integer arrays $a$ and $b$ of size $n$ are **complementary** if there exists an integer $x$ such that $a_i + b_i = x$ over all $1 \le i \le n$. For example, the arrays $a = [2, 1, 4]$ and $b = [3, 4, 1]$ are complementary, since $a_i + b_i = 5$ over all $1 \le i \le 3$.

You are given two integer arrays $a$ and $b$. It is known that $a$ and $b$ both contain $n$ non-negative integers not greater than $k$. Unfortunately, some elements in $b$ are lost, denoted with $-1$. Help count the number of possible arrays $b$ such that $a$ and $b$ are complementary and all lost elements are replaced with non-negative integers no more than $k$.

## STAGE 1: PLANNER OUTPUT

### Planner JSON Output

```
{
  "rationale": "To ensure complementarity across known and unknown
      ↪ positions, we derive a candidate sum x from known pairs and
      ↪ count how many values in unknown positions would satisfy bi
      ↪ = x - ai within bounds.",
  "plan": [
    "Decomposition",
    "FormalDeduction",
    "AnalogicalReasoning",
    "AlgorithmicImplementation",
    "CrossVerification",
    "PrincipleGeneralization"
```

```
    ]
}
```

STAGE 2: EXECUTOR OUTPUT

The Executor receives the problem and the plan above and produces the following detailed solution.

**Executing Module: Decomposition**    **Goal:** Determine the number of valid values for a constant sum $x$ such that for all $i$, $a_i + b_i = x$ holds, where some $b_i$ are known, and others are unknown (denoted by -1). Each missing $b_i$ must satisfy $0 \leq b_i \leq k$.

**Inputs:** $t$ (number of test cases). For each test case: $n$ (size of arrays), $k$ (max allowed value), arrays $a$ and $b$.

**Subproblems:**

1. From known positions where $b_i \neq -1$, infer the required sum $x = a_i + b_i$. All known pairs must imply the same value of $x$.

2. For unknown positions where $b_i = -1$, count how many candidate $x$ values allow a valid replacement, i.e., $0 \leq x - a_i \leq k$.

**Executing Module: FormalDeduction**    From all known $(a_i, b_i)$ pairs, derive a candidate value $x = a_i + b_i$. If these pairs imply different sums, then no solution exists (count is 0).

If all known pairs are consistent (or if there are no known pairs), validate that for all missing $b_i$, the required value $x - a_i$ lies in the valid range $[0, k]$.

If there are no known $b_i$ values, then we must explore all possible values for $x$. For each position $i$, the constraint is $0 \leq x - a_i \leq k$, which is equivalent to $a_i \leq x \leq a_i + k$. The set of valid $x$ values is the intersection of these intervals $[a_i, a_i + k]$ across all $i$.

**Executing Module: AnalogicalReasoning**    This problem is analogous to:

- **Constraint satisfaction:** The target sum $x$ must satisfy constraints from multiple sources (the known pairs). It then defines feasible ranges for the unknown variables.

- **Interval intersection:** When $x$ is not fixed by any known pair, the problem becomes finding the size of the intersection of multiple intervals, where each interval $[a_i, a_i + k]$ represents the valid range for $x$ imposed by position $i$.

Final C++ Solution

**Executing Module: AlgorithmicImplementation**

```cpp
#include <iostream>
#include <vector>
#include <algorithm>
#include <climits>

using namespace std;
using ll = long long;

void solve() {
    int n;
    ll k;
    cin >> n >> k;
    vector<ll> a(n), b(n);
    for (ll &x : a) cin >> x;
    for (ll &x : b) cin >> x;

    ll x = -1;
    bool possible = true;
    for (int i = 0; i < n; ++i) {
```

```
918          if (b[i] != -1) {
919              ll candidate = a[i] + b[i];
920              if (x == -1) {
921                  x = candidate;
922              } else if (x != candidate) {
923                  possible = false;
924                  break;
925              }
926          }
927      }
928
929      if (!possible) {
930          cout << 0 << '\n';
931          return;
932      }
933
934      if (x != -1) {
935          // Case 1: x is determined by at least one known pair.
936          for (int i = 0; i < n; ++i) {
937              if (b[i] == -1) {
938                  ll required_b_i = x - a[i];
939                  if (required_b_i < 0 || required_b_i > k) {
940                      possible = false;
941                      break;
942                  }
943              }
944          }
945          cout << (possible ? 1 : 0) << '\n';
946      } else {
947          // Case 2: x is unconstrained by any known pair.
948          ll lo = 0, hi = LLONG_MAX;
949          for (int i = 0; i < n; ++i) {
950              ll left = a[i];
951              ll right = a[i] + k;
952              lo = max(lo, left);
953              hi = min(hi, right);
954          }
955          if (lo > hi) {
956              cout << 0 << '\n';
957          } else {
958              cout << hi - lo + 1 << '\n';
959          }
960      }
961  }
962
963  int main() {
964      ios_base::sync_with_stdio(false);
965      cin_tie(NULL);
966      int t;
967      cin >> t;
968      while (t--) {
969          solve();
970      }
971      return 0;
972  }
```

**Executing Module: CrossVerification  Test Case:** $a = [1, 3, 2], b = [-1, -1, 1], k = 10$. From $a_2, b_2$, we know $x = a_2 + b_2 = 2 + 1 = 3$. Check missing values: For $i = 0$, $b_0 = x - a_0 = 3 - 1 = 2$. Is $0 \le 2 \le 10$? Yes. For $i = 1$, $b_1 = x - a_1 = 3 - 3 = 0$. Is $0 \le 0 \le 10$? Yes. All checks pass. There is exactly 1 way. Output: 1. Correct.

**Edge Case:** All $b_i = -1$. $a = [1, 2], k = 1$. The intervals for $x$ are $[1, 1 + 1] = [1, 2]$ and $[2, 2 + 1] = [2, 3]$. The intersection is $[2, 2]$. The size is $2 - 2 + 1 = 1$. Output: 1. Correct.

**Executing Module: PrincipleGeneralization**   The core principle is to determine the constraints on the complementary sum $x$.

1. If any known pairs $(a_i, b_i)$ exist, they fix the value of $x$. We must first verify that all known pairs are consistent with this single $x$. Then, we verify if this fixed $x$ allows all unknown $b_i$ to be in the range $[0, k]$. If all checks pass, there is 1 way; otherwise, 0.

2. If no known pairs exist, $x$ is constrained by every position. For each position $i$, $x$ must be in $[a_i, a_i + k]$. The total number of ways is the size of the intersection of all these intervals.

The time complexity is $O(n)$ per test case.

### B.2   Example 2: Prefix Min, Suffix Max

> **Problem Statement**
>
> You are given an array $a$ of distinct integers. In one operation, you may either:
>
> - choose a nonempty prefix[a] of $a$ and replace it with its minimum value, or
> - choose a nonempty suffix[b] of $a$ and replace it with its maximum value.
>
> For each element $a_i$, determine if there exists some sequence of operations to transform $a$ into $[a_i]$. Output your answer as a binary string of length $n$.
>
> ---
> [a]A prefix of an array is a subarray consisting of the first $k$ elements of the array, for some integer $k$.
> [b]A suffix of an array is a subarray consisting of the last $k$ elements of the array, for some integer $k$.

### Stage 1: Planner Output

Planner JSON Output

```
{
  "rationale": "Since each target requires shrinking the array via
      ↪ prefix-min or suffix-max operations, we decompose reachable
      ↪ intervals and rigorously derive which elements can remain.",
  "plan": [
    "Decomposition",
    "AnalogicalReasoning",
    "FormalDeduction",
    "AlgorithmicImplementation",
    "CrossVerification",
    "PrincipleGeneralization"
  ]
}
```

### Stage 2: Executor Output

The Executor receives the problem and the plan above and produces the following detailed solution.

**Executing Module: Decomposition   Goal:** For each position $i$ in a distinct-integer array of length $n$, determine if we can sequence prefix-min and suffix-max compressions to end with the single element $a_i$.

**Variables & Inputs:** $n$ (array length), $a[0 \ldots n-1]$ (distinct integers).

**Operation Effects:**

- Prefix-min on prefix $[0 \ldots k]$ replaces that block with its minimum.

- Suffix-max on suffix $[k \dots n-1]$ replaces that block with its maximum.

Each operation reduces the array's length. The final state is an array of length 1.

**Subproblem:** Characterize exactly which elements $a_i$ can become the final surviving value through some sequence of these operations.

**Executing Module: AnalogicalReasoning** This problem has parallels with algorithms that select endpoints or extremal values:

- **Stack-based nearest smaller/larger element:** The logic of an element being a minimum/maximum in a prefix/suffix is related to finding the nearest element that is smaller or larger.
- **Greedy choices:** To make $a_i$ survive, it must be chosen as an extremum (min of a prefix or max of a suffix) at some point. This suggests a greedy logic. An element can survive if it's "stronger" (smaller for prefix-min, larger for suffix-max) than its neighbors on at least one side.

**Executing Module: FormalDeduction** For an element $a_i$ to be the final result, it must survive all operations. Consider the final operation. The array will have some elements $[c_1, c_2, \dots, c_m]$, and we want to perform an operation that results in $[a_i]$. This means $a_i$ must have been one of the elements in $[c_1, \dots, c_m]$.

- If the last operation is a prefix-min on the whole array, then $a_i$ must be the minimum of $[c_1, \dots, c_m]$.
- If the last operation is a suffix-max on the whole array, then $a_i$ must be the maximum of $[c_1, \dots, c_m]$.

Let's trace this back. The elements $[c_1, \dots, c_m]$ are themselves results of previous operations. An element $a_i$ can only ever be created by being the minimum of a prefix of the original array or the maximum of a suffix. Any other element cannot be generated. So, for $a_i$ to survive, it must be possible to shrink the array around it, leaving it as an extremum.

A simpler observation: If we want to end with $a_i$, we can shrink the prefix up to $i-1$ and the suffix from $i+1$. To preserve $a_i$, when we shrink the prefix $[0, \dots, j]$ with $j \geq i$, $a_i$ must be the minimum. Similarly for the suffix. An element $a_i$ can be the final result if we can perform a sequence of operations to eliminate all other elements.

1. We can eliminate everything to the left of $a_i$ by repeatedly taking prefix-min operations. For $a_i$ to survive this, it must be smaller than everything to its left. Thus, $a_i = \min(a_0, \dots, a_i)$.
2. We can eliminate everything to the right of $a_i$ by repeatedly taking suffix-max operations. For $a_i$ to survive this, it must be larger than everything to its right. Thus, $a_i = \max(a_i, \dots, a_{n-1})$.

A key insight is that we can choose which side to shrink first. For $a_i$ to be a possible result, it only needs to be an extremum on one side. If $a_i = \min(a_0, \dots, a_i)$, we can apply a prefix-min operation on $[0, \dots, i]$, turning the array into $[a_i, a_{i+1}, \dots, a_{n-1}]$. Now, we can shrink the new suffix until we are left with just $a_i$. Symmetrically, if $a_i = \max(a_i, \dots, a_{n-1})$, we can shrink the suffix first. **Conclusion:** $a_i$ is a reachable target if and only if ($a_i$ is the minimum of its prefix) OR ($a_i$ is the maximum of its suffix).

Final C++ Solution

**Executing Module: AlgorithmicImplementation**

```
#include <iostream>
#include <vector>
#include <string>
#include <algorithm>
#include <climits>
```

```
1080   using namespace std;
1081
1082   void solve() {
1083       int n;
1084       cin >> n;
1085       vector<int> a(n);
1086       for (int i = 0; i < n; i++) {
1087           cin >> a[i];
1088       }
1089       vector<int> sufMax(n);
1090       sufMax[n - 1] = a[n - 1];
1091       for (int i = n - 2; i >= 0; i--) {
1092           sufMax[i] = max(sufMax[i + 1], a[i]);
1093       }
1094       int preMin = INT_MAX;
1095       string result = "";
1096       for (int i = 0; i < n; i++) {
1097           preMin = min(preMin, a[i]);
1098           bool is_possible = (a[i] == preMin) || (a[i] == sufMax[i]);
1099           result += (is_possible ? '1' : '0');
1100       }
1101       cout << result << "\n";
1102   }
1103
1104   int main() {
1105       ios_base::sync_with_stdio(false);
1106       cin_tie(NULL);
1107       int t;
1108       cin >> t;
1109       while (t--) {
1110           solve();
1111       }
1112       return 0;
1113   }
```

**Executing Module: CrossVerification   Test Case 1:** 'a = [1, 3, 5, 4, 7, 2]' Prefix mins: '[1, 1, 1, 1, 1, 1]' Suffix maxs: '[7, 7, 7, 7, 7, 2] - 'a[0]=1': '1 == preMin[0]'. Yes. -¿ 1 - 'a[1]=3': '3 != preMin[1]', '3 != sufMax[1]'. No. -¿ 0 - 'a[2]=5': '5 != preMin[2]', '5 != sufMax[2]'. No. -¿ 0 - 'a[3]=4': '4 != preMin[3]', '4 != sufMax[3]'. No. -¿ 0 - 'a[4]=7': '7 != preMin[4]', '7 == sufMax[4]'. Yes. -¿ 1 - 'a[5]=2': '2 != preMin[5]', '2 == sufMax[5]'. Yes. -¿ 1 Result: '100011'. Correct.

**Edge Cases:** - Strictly increasing array '[1, 2, 3, 4]': Every element is a prefix minimum. 'preMin[i] == a[i]'. All '1's. - Strictly decreasing array '[4, 3, 2, 1]': Every element is a suffix maximum. 'sufMax[i] == a[i]'. All '1's. The logic holds.

**Executing Module: PrincipleGeneralization**   An element $a_i$ in an array of distinct integers can be isolated as the sole remaining element via prefix-min and suffix-max operations if and only if it is an "extremal" element from its position to one of the array's boundaries. Specifically, $a_i$ must be either the minimum of the prefix $a[0 \ldots i]$ or the maximum of the suffix $a[i \ldots n - 1]$. This can be efficiently checked by pre-computing all prefix minimums and suffix maximums in $O(n)$ time and then checking the condition for each element.

B.3   EXAMPLE 3: VISUAL REASONING (SVG PATH - HEPTAGON)

Problem Statement This SVG path element `<path d="M 55.57,80.69 L 57.38,65.80 M 57.38,65.80 L 48.90,57.46 M 48.90,57.46 L 45.58,47.78 M 45.58,47.78 L 53.25,36.07 L 66.29,48.90 L 78.69,61.09 L 55.57,80.69"/>` draws a

(A) circle

(B) heptagon

(C) hexagon

(D) kite

(E) line

(F) octagon

(G) pentagon

(H) rectangle

(I) sector

(J) triangle

STAGE 1: PLANNER OUTPUT

Planner JSON Output

```
{
  "rationale": "The problem requires interpreting an SVG path as a
      ↪ polygonal shape, which benefits from decomposing coordinates
      ↪ into edges and verifying polygon type through formal
      ↪ geometric reasoning.",
  "plan": [ "Decomposition", "FormalDeduction", "CrossVerification",
      ↪ "PrincipleGeneralization" ]
}
```

STAGE 2: EXECUTOR OUTPUT

**Executing Module: Decomposition   Goal:** Identify the geometric figure from the SVG path. **Variables:** The path consists of M (move) and L (line) commands. **Vertices:** (55.57, 80.69), (57.38, 65.80), (48.90, 57.46), (45.58, 47.78), (53.25, 36.07), (66.29, 48.90), (78.69, 61.09). The path ends by returning to the first vertex. **Sub-problems:** Count distinct vertices and confirm path closure.

**Executing Module: FormalDeduction   Vertex Count:** There are 7 distinct vertices. **Closure:** The final 'L' command returns to the starting point, so the shape is a closed polygon. **Polygon Type:** A polygon with 7 vertices is a heptagon. All other options are eliminated as they have a different number of sides or are not polygons made of straight lines.

**Executing Module: CrossVerification   Segment Count:** Counting the line segments between vertices gives: 1→2, 2→3, 3→4, 4→5, 5→6, 6→7, and 7→1. This is a total of 7 segments (edges). **Consistency Check:** A 7-sided polygon is a heptagon. This is consistent with the deduction.

**Executing Module: PrincipleGeneralization   To** identify a polygon from an SVG path with only M and L commands, parse the coordinates to find the number of unique vertices. If the path is closed, this number corresponds to the number of sides, which defines the type of polygon (e.g., 7 vertices = heptagon).

**Final Answer:** The correct option is (B) heptagon.

B.4   EXAMPLE 4: VISUAL REASONING (SVG PATH - OCTAGON)

Problem Statement This SVG path element `<path d="M 50.55,76.16 L 40.00,72.83 M 40.00,72.83 L 34.22,59.10 M 34.22,59.10 L 38.51,46.47 L 49.40,60.03 L 68.54,44.66 L 72.08,58.87 M 72.08,58.87 L 61.72,73.75 L 50.55,76.16"/>` draws a

(A) circle

(B) heptagon

(C) hexagon

(D) kite

(E) line

(F) octagon

(G) pentagon

(H) rectangle

(I) sector

(J) triangle

STAGE 1: PLANNER OUTPUT

Planner JSON Output

```
{
  "rationale": "The problem requires interpreting a sequence of SVG
      ↪ path coordinates as a closed polygon, so we must break down
      ↪ the path into vertices, deduce the polygon type rigorously,
      ↪ verify correctness, and generalize the approach.",
  "plan": [ "Decomposition", "FormalDeduction", "CrossVerification",
      ↪ "PrincipleGeneralization" ]
}
```

STAGE 2: EXECUTOR OUTPUT

**Executing Module: Decomposition   Goal:** Identify the geometric figure from the SVG path. **Vertices:** (50.55, 76.16), (40.00, 72.83), (34.22, 59.10), (38.51, 46.47), (49.40, 60.03), (68.54, 44.66), (72.08, 58.87), (61.72, 73.75). The path ends by returning to the first vertex. **Sub-problems:** Count distinct vertices and confirm path closure.

**Executing Module: FormalDeduction   Vertex Count:** There are 8 distinct vertices. **Closure:** The final 'L' command returns to the starting point, so the shape is a closed polygon. **Polygon Type:** A polygon with 8 vertices is an octagon. All other options are eliminated.

**Executing Module: CrossVerification   Segment Count:** Counting the line segments between vertices gives: 1→2, 2→3, 3→4, 4→5, 5→6, 6→7, 7→8, and 8→1. This is a total of 8 segments (edges). **Consistency Check:** An 8-sided polygon is an octagon, which is consistent.

**Executing Module: PrincipleGeneralization**   The same principle as the previous example applies: for a closed SVG path composed of straight line segments, the number of unique vertices determines the type of polygon (e.g., 8 vertices = octagon).

**Final Answer:** The correct option is (F) octagon.

