# OpenReview forum: "Dynamic Cognitive Orchestration: Eliciting Metacognitive Planning in Large Language Models"
_ICLR.cc/2026/Conference — ICLR 2026 Conference Withdrawn Submission_

### Official Review · Reviewer_MGna · 2025-11-01

**Soundness:** 2
**Presentation:** 3
**Contribution:** 2
**Rating:** 2
**Confidence:** 4

**Summary:**

This paper proposes a two-stage prompting technique for language model to solve complicated reasoning tasks. The first stage prompts the LM to generate a plan from a predefined set of tools (meta-cognition stage), and the second stage prompts the LM to execute its own plan. Experimental results show superiority over baseline reasoning techniques.

**Strengths:**

The motivation of equipping agents with a mega-cognitive planning stage is well-justified.

**Weaknesses:**

The novelty of the proposed approach is limited, as planning in various forms have been widely adopted in agent design. Also since the planning stage of the proposed method is limited to the items from the predefined toolbox, it remains unclear how comprehensive and generalizable this toolbox is.

**Questions:**

- How are the tools in Table 1 selected? Did it follow any principles from cognitive study? This is one of the core proposals of this paper, yet the design appears to be a bit arbitrary, and the analysis in Sec 6.1 did not cover all uses case of these tools.
- Meta-cognition goes beyond just planning, it also includes qualities like self-reflection and self-improvements. I wonder what if you just apply self-reflection to baseline methods like ToT or Analogical Prompting, would the results also become much better? The baselines being considered could be made much stronger.

---

### Official Review · Reviewer_6ne1 · 2025-11-01

**Soundness:** 2
**Presentation:** 2
**Contribution:** 2
**Rating:** 2
**Confidence:** 5

**Summary:**

**Summary:**
The paper proposes Dynamic Cognitive Orchestrator (DCO), a two-stage prompting framework that brings metacognitive planning into LLM problem solving. In this framework, the Planner first analyzes the problem and generates a customized reasoning plan by selecting from a predefined *toolbox of cognitive modules*, while the Executor follows this plan step-by-step to derive the final answer. This design aims to overcome the rigidity of static prompting methods such as Chain-of-Thought and Tree-of-Thought by enabling adaptive, problem-dependent reasoning.

**Contributions:**
(1) Proposes a Planner–Executor framework that explicitly separates high-level reasoning planning from execution.
(2) Designs a set of atomic cognitive strategies and corresponding prompt templates for both the Planner and the Executor, providing a practical implementation of dynamic reasoning orchestration.

**Strengths:**

**Originality:** The paper proposes a Planner–Executor framework and provides a simple implementation through prompt engineering.

**Quality:** The paper is clearly written and easy to follow.

**Experiments:** Results on several benchmarks demonstrate the effectiveness of the approach, and the analysis shows that different cognitive modules are optimal for different problem types, highlighting the necessity of dynamic module planning.

**Weaknesses:**

1. **Limited originality**: The overall originality of the paper is moderate. Both the problem of relying on a fixed prompting strategy and the proposed planner–executor framework have been explored in previous works [1–4]. The paper does not clearly articulate its distinctive contributions or advantages over these similar approaches.

2. **Insufficient comparison with related work**: Several relevant studies that also adopt planner–executor architectures [1–4] are not adequately discussed or compared. A more thorough review and positioning of the proposed method within this line of research would enhance the paper’s contextual depth.

3. **Lack of ablation study on designed cognitive modules**: The paper does not include an ablation analysis to evaluate the individual effectiveness and necessity of the designed cognitive modules.

4. **Incomplete analysis of strategy usage**: While Table 4 presents the distribution of individual cognitive modules across datasets, the analysis would benefit from an additional distribution analysis of complete generated plan (i.e., the planned array of multiple cognitive modules), which are crucial to understanding the behavior and dynamics of the proposed planner–executor framework.

5. **Increased inference cost**: The proposed approach likely introduces additional computational overhead. A comparative analysis of inference time and API cost would strengthen the empirical evaluation and provide a clearer picture of the method’s practical trade-offs.

**References**

[1] *Chain of Methodologies: Scaling Test Time Computation without Training* ([https://aclanthology.org/2025.findings-acl.276/](https://aclanthology.org/2025.findings-acl.276/))

[2] *rStar: Mutual Reasoning Makes Smaller LLMs Stronger Problem-Solver* ([https://openreview.net/pdf?id=6aHUmotXaw](https://openreview.net/pdf?id=6aHUmotXaw))

[3] *DOTS: Learning to Reason Dynamically in LLMs via Optimal Reasoning Trajectories Search* ([https://openreview.net/pdf?id=tn2mjzjSyR](https://openreview.net/pdf?id=tn2mjzjSyR))

[4] *HiAR-ICL: Beyond Examples – High-level Automated Reasoning Paradigm in In-Context Learning via MCTS* ([https://arxiv.org/pdf/2411.18478](https://arxiv.org/pdf/2411.18478))

**Questions:**

Additional questions apart from those listed in **Weaknesses**:

1. **Planner–Executor Separation**: Why is an explicit separation between the planner and executor necessary? Long-CoT models (e.g., o1, R1) already demonstrate meta-cognitive planning abilities, including decomposition and self-reflection, without relying on manually defined action sets. What advantages does the proposed framework offer compared to these models?

2. **Cognitive Module Toolbox Design**: How were the high-level strategies designed, and to what extent are they general and complete?

3. **Experimental Clarity**: In Table 2, the note “Baseline results are from original papers or reproduced for comparability” is mentioned, but it is unclear which results are cited and which are reproduced. In Table 2, the zero-shot CoT performance appears unusually low—strong models like GPT-4o should not exhibit such a large gap from few-shot CoT, given their inherent chain-of-thought capability. Could the authors confirm whether this discrepancy arises from answer extraction issues rather than genuine reasoning failures?

---

### Official Review · Reviewer_YfXy · 2025-11-02

**Soundness:** 1
**Presentation:** 1
**Contribution:** 1
**Rating:** 0
**Confidence:** 5

**Summary:**

The paper proposes Dynamic Cognitive Orchestrator (DCO), a two-stage prompting framework where an LLM first plans a reasoning strategy from a set of cognitive modules and then executes that plan. The authors argue this separates metacognition from problem solving and better mirrors human executive control. They report strong results on MATH, Codeforces, and BIG-Bench Hard and claim that adaptive module selection (e.g., picking “FormalDeduction” for algebra) is what drives the gains.

**Strengths:**

The paper propose a two-stage promping framework and run some experiments.

**Weaknesses:**

1.	It is not clear whether the reported improvements actually stem from the proposed two-stage DCO separation, rather than from extra guidance/token budget or task-specific module design.
2.	The existing reasoning model can already follow a “plan-then-execute” pipeline; the paper should clarify what is fundamentally new in DCO beyond this existing paradigm.
3.	The experimental section is relatively thin (only three tables in an eight-page paper); more comprehensive evaluations and ablations are needed to convincingly justify the method.

**Questions:**

Please check the weakness part.

---

### Official Review · Reviewer_rTAj · 2025-11-04

**Soundness:** 2
**Presentation:** 3
**Contribution:** 2
**Rating:** 2
**Confidence:** 3

**Summary:**

This paper introduces the Dynamic Cognitive Orchestrator (DCO), a two-stage prompting framework designed to make LLMs more adaptive in their reasoning, similar to human cognition.

The authors argue that existing methods like CoT or ToT enforce a fixed, rigid reasoning strategy. In contrast, DCO works in two steps:
- Planner Stage: The LLM first analyzes a problem and creates a bespoke, problem-specific reasoning plan by selecting from a "toolbox" of cognitive modules (e.g., 'Decomposition', 'FormalDeduction', 'CrossVerification').
- Executor Stage: The LLM then receives its own plan and systematically executes it to derive the final solution.

By separating high-level planning from execution, DCO elicits metacognitive behavior, compelling the model to first "reason about how to reason." The paper demonstrates that this approach achieves great results on challenging benchmarks, including 89.2% on MATH, 42.0% on Codeforces, and 89.5% on BIG-Bench Hard, outperforming previous methods.

**Strengths:**

- This paper introduces a conceptually clear two-stage framework (Planner-Executor) that elicits metacognitive reasoning in LLMs.
- Its design is well-grounded in cognitive science, explicitly modeling the brain's executive functions of planning and execution, which provides a strong theoretical justification.
- The paper provides compelling analysis showing that the model dynamically adapts its strategy to the problem type, such as using formal deduction for algebra and heuristics for geometry.
- The explicit generation of a reasoning plan enhances the interpretability of the model's process and makes it highly effective for interactive refinement and error correction.

**Weaknesses:**

- The "Cognitive Module Toolbox" is a fixed, human-engineered constraint that limits the model's autonomy and introduces significant bias. The framework's success is entirely dependent on the researchers' foresight in creating a relevant and comprehensive set of modules, which undermines the claim of truly dynamic and adaptive reasoning.

- The entire evaluation relies on a single, top-tier proprietary model (GPT-4o). This makes it impossible to know if the DCO framework is a genuinely novel and generalizable architecture or simply an elaborate prompt engineering technique that happens to work well with one specific, highly capable model. The results may not replicate on other LLMs.

- The separation between planning and execution is artificially rigid. In reality, expert human reasoning involves continuously reassessing a plan during execution. DCO's Executor cannot adapt or request a new plan if it encounters an unforeseen obstacle, a key limitation the paper itself hints at as future work.

- The performance improvements, while achieving state-of-the-art, are incremental (e.g., 3.6% over Tree-of-Thoughts on MATH). This raises the question of whether the added complexity and latency of a two-stage API call process are justified for what amounts to a single-digit percentage gain over other advanced prompting methods.

- The evaluation benchmarks, while challenging, are confined to analytical and algorithmic reasoning. The paper's claims of emulating general human cognition are not tested against creative, social, or open-ended reasoning tasks where the predefined cognitive modules would likely be insufficient.

**Questions:**

See weaknesses.

---

### Note · Authors · 2025-11-14

I have read and agree with the venue's withdrawal policy on behalf of myself and my co-authors.